# Moral distress measurement in animal care workers: a systematic review

Yigit Baysal ®,[1] Nastassja Goy,[2] Sonja Hartnack ®,[1] Irina Guseva Canu ®[2]

YB and NG contributed equally.

¹Section of Epidemiology, University of Zurich Vetsuisse Faculty, Zurich, Switzerland
²Department of Occupational and Environmental Health, Center for Primary Care and Public Health (Unisanté), University of Lausanne, Lausanne, Switzerland

**Correspondence to**
Yigit Baysal; yigit.baysal@uzh.ch

## ABSTRACT

**Objectives** The mental health of veterinary and other animal health professionals is significantly impacted by the psychological stressors they encounter, such as euthanasia, witnessing animal suffering and moral distress. Moral distress, initially identified in nursing, arises when individuals are aware of the right action but are hindered by institutional constraints. We aimed to review existing research on moral distress scales among animal care workers by focusing on the identification and psychometric validity of its measurement.

**Design** Two-step systematic review. First, we identified all moral distress scales used in animal care research in the eligible original studies. Second, we evaluated their psychometric validity, emphasising content validity, which is a critical aspect of patient-reported outcome measures (PROMs). This evaluation adhered to the Consensus-based Standards for the Selection of Health Measurement Instruments (COSMIN). The results were reported according to the Preferred Reporting Items for Systematic Reviews and Meta-Analyses.

**Data sources** PubMed, EMBASE and PsycINFO to search for eligible studies published between January 1984 and April 2023.

**Eligibility criteria for selecting studies** We included original (primary) studies that (1) were conducted in animal care workers; (2) describing either the development of a moral distress scale, or validation of a moral distress scale in its original or modified version, to assess at least one of the psychometric properties mentioned in COSMIN guidelines.

**Data extraction and synthesis** Two independent reviewers used standardised methods to search, screen and code included studies. We considered the following information relevant for extraction: study reference, name and reference of the moral distress scale used, psychometric properties assessed and methods and results of their assessments. The collected information was then summarised in a narrative synthesis.

**Results** The review identified only one PROM specifically adapted for veterinary contexts: the Measure of Moral Distress for Animal Professionals (MMD-AP), derived from the Measure of Moral Distress for Healthcare Professionals (MMD-HP). Both MMD-HP and MMD-AP were evaluated for the quality of development and content validity. The development quality of both measures was deemed doubtful. According to COSMIN, MMD-HP's content validity was rated as sufficient, whereas MMD-AP's was inconsistent. However, the evidence quality for both PROMs was rated low.

**Conclusion** This is the first systematic review focused on moral distress measurement in animal care workers.

## STRENGTHS AND LIMITATIONS OF THIS STUDY

⇒ The adherence to Consensus-based Standards for the Selection of Health Measurement Instruments guidelines for assessing the content validity of patient-reported outcome measures ensures rigorous evaluation of the scales of moral distress (MD) and their psychometric validity.

⇒ The strength of the review's conclusions is inherently dependent on the quality and robustness of the original studies included.

⇒ MD was coined in 1984 in the nursing discipline and since then further definitions have been developed, possibly leading to a heterogeneous and complex understanding of MD.

It shows that moral distress is rarely measured using standardised and evidence-based methods and that such methods should be developed and validated in the context of animal care.

**PROSPERO registration number** CRD42023422259.

## BACKGROUND

### Animal care workers' mental health and work stressors

The WHO defines mental health as a state of well-being essential for coping with daily life and engaging in meaningful work, beyond just the absence of mental disorders.[1] The Global Burden of Disease study highlights a bidirectional relationship between work and mental health.[2] Participating in meaningful work serves as a protective factor for mental health, while mental well-being influences occupational functioning and quality of professional life.[1] However, work-related stressors, particularly those with moral challenges, can negatively impact mental health.[3] Occupational mental health issues like burnout, compassion fatigue and moral distress (MD) have been studied in healthcare workers but less so in other professions facing similar stressors, such as animal care workers.[4–6] Animal care workers, including laboratory technicians and veterinarians, are at risk of mental ill health and suicide, particularly due to unique stressors like euthanasia.[4 5 7–9] Compared with

human healthcare professionals, research on MD among animal care workers is limited.

Both human and animal healthcare professionals, directly or indirectly being responsible for the care of their patients, form emotional bonds with them. However, at the same time, they may perform procedures that may cause pain and distress to their patients or find themselves unable to alleviate their pain and distress. Occupational mental health of animal care professionals has recently been emphasised, that is, LaFollette *et al* suggested that animal care professionals are significantly experiencing emotional dissonance and moral stress.[10] A key difference between human healthcare and animal care professionals regarding workplace stress lies in the outcomes within clinical and experimental settings. The endpoint in animal experimentation is typically the death of the animal. In harm-benefit analysis it is crucially important that attempts are made to minimise harm and suffering in animal experimental setting. This ethical framework is often in conflict with the predisposition of attachment, love and empathy based on human–animal bonds. Animal care workers often find themselves constantly making and breaking these emotional bonds.[10] Combined with the role conflict experienced due to balancing competing interests of animal welfare and human interests, moral dilemma experienced at the workplace can transform into MD.[8] Social support networks can buffer the effects of moral dilemma and role conflict by providing emotional support and alternative perspectives that help the individual navigate ethical dilemmas and role demands. Yet, a systematic review of workplace stress in animal care workers found reports of qualitative studies where stigma associated with animal related work tasks (eg, euthanasia, use of animals in research) can hinder the access of social support networks.[11] Considering the complexity of MD in animal care work, we emphasise the need for exploration of this psychological syndrome targeted to animal care workers.

Moreover, experiencing MD and its negative consequences potentially impacts animal welfare, including the implementation of the 3R principle (replacement, reduction, refinement) in the animal experimental setting with animal care takers being responsible for fostering these.[12] An association between poorer quality of life in laboratory animal personnel and limited control over euthanasia procedures was found.[10]

## Concept of MD and its attributes

MD was first described in nursing as 'when one knows the right thing to do, but institutional constraints make it nearly impossible to pursue the right course of action'.[13 14] It involves complicity in wrongdoing, lack of voice and conflict with professional values, often exacerbated in hierarchical work environments.[13–15] In veterinary medicine, MD arises from practices like euthanasia and witnessing animal suffering, with personal experiences and attitudes towards animals influencing its development. Veterinarians also report high levels of compassion fatigue, anxiety,

burnout, depression and suicidal thoughts.[7–9] MD affects both professional and personal quality of life and can exacerbate the Post-Traumatic Stress Disorde (PTSD), burnout and compassion fatigue.[9 16 17]

Despite several systematic reviews on MD in medical workers, none have specifically addressed animal care workers.[18] Various tools, mostly self-administered questionnaires, or patient-reported outcome measures (PROMs)[19] have been developed to measure MD. Front of such diversity, a question of their relevance and validity in the context of animal care practice arises.

### Study objectives

In this study, we aimed to systematically review the available literature to answer two complementary research questions: (1) Which scales or PROMs are used in animal healthcare workers to assess MD? (2) What is the psychometric quality of these PROMs? The quality of a PROM is defined by its validity, reliability and responsiveness, with the content validity being the most important psychometric property.[19] According to the Consensus-based Standards for the Selection of Health Measurement Instruments (COSMIN),[19–21] the content validity is 'the degree to which the content of a scale is an adequate reflection of the construct to be measured'.[20] Thus, we focused on the content validity, while assessing the quality of the identified MD measures.

## METHODS

The study involved two steps: initially, a systematic review to identify MD scales conducted in animal care worker. By scale, we understand any standardised questionnaire, some of which could be a PROM. PROM is a standardised, validated questionnaire/rating scale that is completed by patients to capture their perceptions of their health, exposure and quality of life.[22] The second step focused on assessing the psychometric validity of these PROMs, as per COSMIN guidelines.[23] The results were reported according to the Preferred Reporting Items for Systematic Reviews and Meta-Analyses (PRISMA)[24] and COSMIN guidelines.[23]

### Step 1. Systematic review of the MD measurements used in animal care workers

The study protocol was conducted according to protocol registered in Prospective Register of Systematic Reviews (PROSPERO) under the registration number CRD42023422259. We pre-registered this review to specify the research plan and design, to ensure discoverability of the study progress and to report the results systematically. By pre-registering this review, we wanted to overcome our confirmation and hindsight biases to the topic.[25]

### Eligibility criteria

In this review, we included the studies meeting the following criteria: (1) original (primary) study conducted in animal care workers, (2) describing either the development

of an MD measure or validation of an MD measure in its original or modified version and (3) published in English, German, French, Turkish or Russian languages. We excluded: (1) reviews, editorials, commentaries and conference papers, (2) studies using or assessing other outcomes than MD and (3) grey literature.

### Data sources and search terms

We used three databases (PubMed, EMBASE and PsycINFO) to search for eligible studies published between January 1984 and April 2023. In the initial phase of planning our study, we considered reviewing literature over an arbitrary but long period of 50 years. However, on discovering the term 'moral distress' first coined by Jameton in 1984, we adjusted our review period to start from that year onwards. An experienced librarian developed the research strategy that consisted of free-text words to define three search strings, the terms focusing on the population of interest (ie, animal care workers), terms related to MD and terms related to PROMs. The latter was facilitated using the application of a sensitive filter developed by Terwee *et al*.[20] Reference lists were manually checked to identify additional studies during the second screening of full texts of included studies.

### Study screening and selection

The librarian imported all items retrieved from each of the three databases into the bibliography software EndNote V.20 and removed the duplicates. The items were then imported into the Rayyan application[26] for screening. The screening of eligible studies was conducted by two independent reviewers (YB, NG) in two steps. The first screening was conducted based on the title and abstract according to the criteria described above. The second screening was based on full text reading where the same inclusion criteria were applied. All eligible or unclassifiable items were included in the second screening, conducted based on the full-text reading. In both steps, reviewers discussed the discrepancies, and when necessary, they consulted a third reviewer (IGC), as suggested by COSMIN guidelines.[20] As part of the quality control, 25% of the items in both steps of screening were cross-checked by a fourth independent reviewer with expertise in veterinary research (SH).

### Step 2. Systematic review of psychometric validity of the identified MD scales

In the second step of the systematic review, measurement properties of PROMs are assessed using the COSMIN checklist for methodological quality of studies on measurement properties.[21] Assessing the methodological quality of PROMs is essential to determine the adequacy of the use of a PROM in a given target population. The COSMIN checklist outlines the necessary design standards and statistical approaches and allows selection of best PROM by comparing design requirements, sound methodological steps taken and reporting of studies that develop and validate PROMs.[21] The checklist is made up

from the taxonomy of main measurement properties of validity, reliability and responsiveness which includes 12 submeasurement properties.[21] The foremost step in evaluating the adequacy of PROMs according to the COSMIN methodology is content validity assessment since it directly examines whether the instrument reflects the specific domain of interest from the perspective of the target population.[20] Content validity property confirms the ability of the PROMs to be an adequate reflection of the construct they measure. This review was conducted according to the COSMIN guidelines for systematic reviews of PROMs[20 21 23] and the practical guidelines on the identification of the most valid PROM.[27] These guidelines suggest an ordering to evaluate measurement properties of PROMs, starting with content validity. As PROMs base their scale items on reported outcomes of a specific target population, a certain level of confidence in the relevance, completeness and comprehensibility of the PROM items in relation to the construct of interest and the target population needs to be clarified.[19] Later, pooled scores of evaluations of all measurement properties are adjusted based on their quality of evidence using the modified Grading of Recommendations, Assessment, Development, and Evaluations (GRADE) approach for risk of bias.[19 23]

Before evaluating any psychometric properties, the user manual[19] recommends searching relevant PROMs in the COSMIN database of systematic reviews of MD scales in the target population to overcome redundancy. We searched in the COSMIN database for the systematic reviews on MD scales in our target population.[20]

### Data extraction and synthesis

We considered the following information relevant for extraction: study reference, name of the MD scale used, psychometric properties assessed and methods and results of their assessments. Whenever necessary the reviewers contacted the study authors to complete this information. The collected information was then summarised in a narrative synthesis.

### Patient and public involvement

Patients or the public were not involved in the design, or conduct, or reporting or dissemination plans of our research.

## RESULTS

### Included studies

#### Step 1

The literature search resulted in 560 items. After duplicate removal, 455 records were retained for the first screening based on titles and abstracts. For the second screening based on the full-text screening, we selected 16 articles, from which only one met the inclusion criteria (figure 1). In the other articles, either the MD was not measured (n=13), or it was an ineligible type of publication: such as a dissertation (n=1), or the article did not

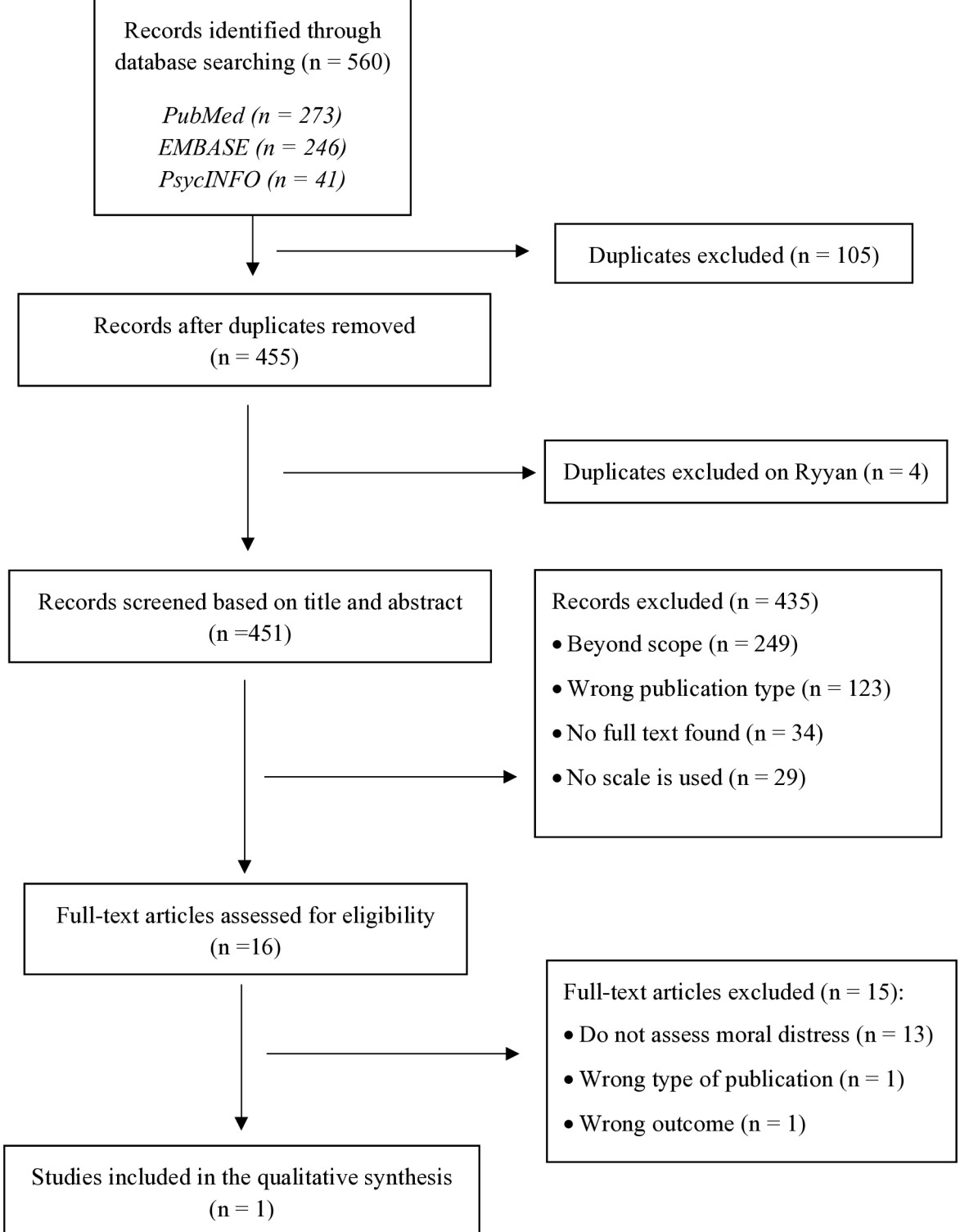

**Figure 1** Flow chart of study selection.

satisfy the inclusion criteria (n=1), only using the MD prevalence as one of the covariates.

The included study was published in 2023 by Kogan and Rishniw[15] and aimed to assess MD in veterinarians, differentiating between the owners of veterinary practices and the employed associates. Kogan and Rishniw found that gender and age were predictors of MD. Indeed, the level of MD was higher in young women than in older men. Moreover, the authors found that a high level of MD was associated with a low level of professional fulfilment and with a high level of exhaustion, burnout and interpersonal disengagement.[15]

### Step 2
In the second step, we evaluated the included studies based on the COSMIN methodology checklist.[21] Following

the COSMIN approach, no systematic review on animal healthcare taker was found in the COSMIN database, but an existing systematic review for human healthcare professionals assessed 32 MD scales in their original, revised and translated versions.[18] Our current systematic review and the previous systematic review by Giannetta *et al*[18] have certain similarities in their research questions, objectives and study design. In both of our research questions/objectives, we tried to identify measures of MD and evaluate them according to COSMIN methodology. Both of the reviews were conducted in a two-step systematic review procedure. In both of the reviews, adaptations of the original studies were identified and evaluated separately.

Nevertheless, despite Giannetta *et al*'s claim of adhering to COSMIN guidelines in their review, the evaluation of the content validity of existing MD PROMs was not thoroughly executed. Therefore, three reviewers (YB, NG, IGC) took it on themselves to assess the content validity of the PROMs identified in the initial step. This fortuitous discovery within Giannetta *et al*'s review, however, led us to the original PROM (Measure of Moral Distress for Healthcare Professionals, MMD-HP),[28] from which our chosen scale, Measure of Moral Distress for Animal Professionals (MMD-AP),[15] was derived. By assessing the original PROM (MMD-HP) along with the adapted version (MMD-AP), we were able to better capture the content of originally selected items and assess their relevance and comprehensiveness with respect to the theoretical foundations of MD construct features.

In the COSMIN framework, all revised versions of a PROM are considered as new PROMs and should be assessed independently. As PROMs measure constructs that can only be reported by the patients themselves, no gold standards exist for these measures.[19] In cases when PROM is adapted to be used in another target population than the original target population for which it was developed, a new evaluation is necessary to appraise the relevancy of items in the original PROM.[21] As an example, a PROM that is originally developed for MD in human healthcare professionals may not be relevant for veterinary professionals. Human healthcare and veterinary medicine may experience different moral dilemmas and constraints, and deal with different stakeholders. While an item may reflect MD related to 'convenience euthanasia' due to animal owner's request, an original PROM from human healthcare would not include this potentially relevant item.

Therefore, in the second step of this study, we considered both PROMs: MMD-HP and MMD-AP for owners of veterinary practices and the employed associates. We chose the MMD-AP version of associates over the practice owners because the associates' version is the same as the owners' version but contains three more items. These items are related to superiors' pressure and their lack of support. Thus, we think this version is more representative of the animal workers' job experience and covers all items in both the MMD-AP versions. Consequently,

we concluded that since no qualitative studies were conducted for item selection of these adapted scales in two different populations, there is no justification for a separate assessment.

## Construct definition and PROM description

The two identified PROMs share the same construct of MD, initially defined by Jameton.[13] Epstein *et al*[28] defined it later as follows: '*The pressure to act unethically is the defining concept of the phenomenon and it separates those situations that are emotionally distressing or otherwise morally troubling from those that threaten moral integrity*'. Epstein *et al*[28] considered five important factors as causes of MD that inspired the MMD-HP structure: complicity in wrongdoing, lack of voice, professional values, the repetitive nature of MD and the three roots of causes. These three roots were ordered at three different levels: patients or their families, for example, when the family asks for a treatment that does more harm to the patient than good. The colleagues or the team can be a cause of MD, for example, in the case of poor communication between team members. Finally, the system can cause MD when there is a lack of human, material or financial resources. MD can be a combination of stressors arising at these three levels. For instance, if the worker faces pressure situations in which he/she cannot report an error committed by a colleague frequently, his/her opinion cannot be considered and his/her professional ethic could be violated.[28]

### Measure of Moral Distress for Healthcare Professionals

The first identified PROM, MMD-HP, was developed in 2019 by revising the Moral Distress Scale-Revised (MDS-R).[29] Epstein *et al* revised MDS-R to make it applicable to all health workers. MDS-R itself originally stems from the Moral Distress Scale (MDS).[30] Epstein *et al* underwent the construction of a new PROM to amend the drawbacks of the original and previous revised versions, namely to simplify the use and to better integrate the causal levels of MD, more specifically the team and system levels. Epstein *et al* reviewed the literature over the past 5 years to identify new roots of causes of MD. The new elements led to the deletion of three items, the modification of three other items, the consolidation of two items and the addition of 11 new items. The new items integrated the team and system levels, which, according to Epstein *et al*, were a weakness of MDS-R, mostly focused on the patient level. The MMD-HP contains 27 items and the three levels of roots to separate the items into four factors. Some items appear in more than one factor. The first factor corresponds to the primarily system level, while the second factor corresponds to the clinical level at the patient level. Factors 3 and 4 both represent the team level. However, factor 3 contains items, which imply a threat to personal integrity from a member of his team, whereas factor 4 concerns the team's communication with patients and families.

The MMD-HP intends a double scaling, one for the frequency of the described situations and one for the

degree of distress each situation results in. Both are scored on a 5-point Likert scale, ranging from 0 (never) to 4 (very frequently) for the frequency and from 0 (none) to 4 (very distressing) for the degree of distress. The product of the frequency and the distress scores corresponds to an overall score of MD, ranging from 0 to 432. The higher the score, the greater the level of MD. However, the authors did not mention any cut-off values.[28]

### Measure of Moral Distress for Animal Professionals

The second PROM, MMD-AP, resulted from an additional revision of the MMD-HP. Kogan and Rishniw (14) used the MMD-HP[28] to assess MD in veterinarians. The associates' version, for employed veterinarians, includes 24 items, and the owners' version has 21 items. The authors explained that they removed items which did not fit the veterinary context. The authors considered the five components at three levels, described by Epstein *et al*,[28] which were included in the MMD-AP[15]; these components are complicity in wrongdoing, lack of voice, wrongdoing associated with professional (not personal) values, repeated experiences and three levels of root causes (at patient, unit, system levels).[28] For example, for the root causes system level, one item reads: '*Experience lack of administrative action or support for a problem that is compromising patient care*'.[15] For the patient-level root causes, an item reads: '*Euthanasia based on client's unwillingness to treat*'. For the team level, there is an item: '*Feel pressured by superiors to order or carry out orders for what I consider to be unnecessary or inappropriate tests and treatments*'.[15] For each item, the response scale is based on the frequency and the distress experienced. Response options and calculation of the MD overall score are the same as the MMD-HP.

### Quality of psychometric validity of the selected PROMs
### Quality of the PROM development

The COSMIN guidelines for evaluating the quality of the study on the PROM development consist of two parts. The first part deals with the standards used to assess the quality of the searches carried out to find the most suitable items for the new PROM. It includes general design requirements and concept elicitation. The concept of elicitation is important for relevance and comprehensiveness of the PROM. The second part looks at the standards used to assess the quality of the cognitive interview study or pilot studies conducted. The aim is to rate the comprehensiveness and comprehensibility of the PROM.[20]

For both PROMs, the construct and its origin, the target population and the context of use were well defined and rated as very good. Both PROMs have been developed in English and tested in a sample representing the target population, thus the general design requirements were also rated as very good. We appraised the concept elicitation as doubtful for MMD-AP and adequate for MMD-HP. For these two PROMs, we could not rate them as very good, because the qualitative data collection was not or not enough described, even after contacting the authors for further information on the analyses they carried

out. The total PROM design was thus rated as adequate for MMD-HP and doubtful for MMD-AP. The MMD-AP authors conducted a very good cognitive interview asking the target population (ie, veterinarians) for guidance and feedback, thus we evaluated the general design requirements as very good. For MMD-HP, the rating was adequate because the cognitive interview was performed with experts and clinicians but not with all health professionals. However, Kogan and Rishniw did not ask the participants about the comprehensibility of the items included in the PROM. Comprehensibility is the standard that ensures that PROM is comprehensible to the target population. For a very good rating, different PROM components need to be evaluated separately: instruction, item, response option and recall period.[20] Since the authors did not mention whether they have evaluated these components for MMD-AP, we rated the quality of the comprehensibility study as doubtful. Finally, the total quality of the MMD-AP development study was rated as doubtful. For MMD-HP, the comprehensibility was also assessed as doubtful. Although its authors carried out a study to evaluate the face validity, they did not mention whether they used an interview guide and did verbatim transcriptions.

Table 1 summarises the results of the PROM development assessment, with the items considered and their consensual rating by two independent reviewers.

### Quality of the PROM content validity

The criteria to assess content validity are relevance, comprehensiveness and comprehensibility. The relevance means, for example, ensuring that PROM items are adapted to the target population and to the context in which they are used. The comprehensiveness evaluates the completeness of the PROM and the comprehensibility concerns the understanding of the PROM by the target population. Table 2 summarises the results of these three criteria for PROM content validity assessment.

Quality of evidence for MD scales is assessed using the modified GRADE approach for risk of bias.[19 23] This approach allows reviewers to adjust their evaluations of pooled results of PROM measurement property evaluations. It assumes that pooled results of individual measurement properties have an overestimation of the quality of evidence; thus, they have to be downgraded, given suspicion over trustworthiness of the published results. It consists of five factors to determine the quality of the evidence: risk of bias, inconsistency, indirectness, imprecision and publication bias. Within these factors, three of these factors are applicable to content validity evaluation: risk of bias, inconsistency and indirectness.[19 20] The risk of bias depends on the number of publications available in the literature on a given PROM, from its development to numerous studies evaluating its measurement properties, and the evaluators' judgements based on their appropriateness. For example, the indirectness factor is related to whether qualitative methods are carried out on a representative sample of the intended target population. The

**Table 1** Quality of the PROM development

| PROM | MMD-HP | MMD-AP |
|---|---|---|
| First author, year | Esptein, 2019[28] | Kogan, 2023[15] |
| Language in which the PROM was developed | English | English |
| PROM design | | |
| Clear construct | V | V |
| Clear origin of construct | V | V |
| Clear target population for which the PROM was developed | V | V |
| Clear context of use | V | V |
| PROM developed in sample representing the target population | V | V |
| Concept elicitation | A | D |
| Total PROM design | A | D |
| Cognitive interview study performed in sample representing the target population | A | V |
| Comprehensibility | D | D |
| Total cognitive interview study | D | D |
| Total PROM development | D | D |

A, adequate; D, doubtful; MMD-AP, Measure of Moral Distress for Animal Professionals; MMD-HP, Measure of Moral Distress for Healthcare Professionals; PROM, patient-reported outcome measure; V, very good.

inconsistency factor depends on the variability of ratings given to each item by reviewers based on the COSMIN content validity methodology checklist. If there is no explanation found to explain the variability of ratings, the rating is downgraded to 'inconsistent'.[19]

Table 3 summarises the results of the PROMs overall rating of content validity, with the adjusted quality of evidence.

For the MMD-AP content validity assessment, the development study was evaluated as inconsistent. The lack of information on qualitative data collection for the development of this PROM led to an insufficient assessment of the relevance. For comprehensiveness, the key concepts were missing, and we rated it as insufficient. The comprehensibility was also rated as insufficient because the authors did not report whether they tested or not the understanding of the PROM. Both reviewers rated the development study of this PROM as sufficient. The reviewers estimated that the PROM itself contained the necessary information. Consequently, the overall rating of the MMD-AP was inconsistent. There was no content validity study available for this PROM.[15] Therefore, the lack of studies on content validity and the doubtful PROM development study has resulted in a low quality of evidence.

The content validity of MMD-HP was rated as sufficient. Indeed, the relevance, comprehensiveness and comprehensibility were rated as sufficient, and the two reviewers

**Table 2** Systematic results for the MMD-HP and MMD-AP PROMs for the criteria content validity

| PROM | Development study | Rating of reviewers | Overall rating | Quality of evidence |
|---|---|---|---|---|
| MMD-HP | | | | |
| Relevance | + | ± | + | Low |
| Comprehensiveness | + | + | + | Low |
| Comprehensibility | + | + | + | Low |
| Content validity rating | + | + | + | Low |
| MMD-AP | | | | |
| Relevance | ± | + | ± | Low |
| Comprehensiveness | – | + | ± | Low |
| Comprehensibility | – | + | ± | Low |
| Content validity rating | ± | + | ± | Low |

±, inconsistent; +, sufficient; –, insufficient; MMD-AP, Measure of Moral Distress for Animal Professionals; MMD-HP, Measure of Moral Distress for Healthcare Professionals; PROM, patient-reported outcome measure.

**Table 3** Systematic results for the overall rating of MMD-HP and MMD-AP PROMs and their quality of evidence

| PROM | MMD-HP | | MMD-AP | |
|---|---|---|---|---|
| COSMIN psychometric properties for content validity | Overall rating | Quality of evidence | Overall rating | Quality of evidence |
| Relevance | + | Low | ± | Low |
| Comprehensiveness | + | Low | − | Low |
| Comprehensibility | + | Low | − | Low |

−, insufficient; +, sufficient; ±, inconsistent; COSMIN, Consensus-based Standards for the Selection of Health Measurement Instruments; MMD-AP, Measure of Moral Distress for Animal Professionals; MMD-HP, Measure of Moral Distress for Healthcare Professionals; PROM, patient-reported outcome measure.

also evaluated these properties as sufficient. Overall, the content validity rating was also sufficient. Since we did not find any content validity study for MMD-HP, the quality of evidence grading started as moderate. Then, we downgraded it to low because the PROM's development study has been assessed as doubtful.

### Other psychometric properties of the selected PROMs

According to the COSMIN framework, in case the PROM does not meet the criteria for a sufficient quality of content validity, the assessment of other psychometric properties is useless, the content validity being a paramount requirement for recommending a PROM for use in research or clinical practice. The content validity was inconsistent for MDD-AP and sufficient for MDD-HP; therefore, we did not conduct any formal assessment of the other properties of these PROMs. Nevertheless, Giannetta *et al* made a systematic review of studies assessing additional psychometric properties of MDD-HP.[18] The internal consistency of MDD-HP was graded as sufficient, with a Cronbach's alpha of 0.93 on average.[18] The structural validity was appraised as sufficient. It was assessed by an exploratory factor analysis, where the four identified factors accounted for 54.3% of the variance.[18] Nonetheless, no confirmatory factor analysis was performed, contrary to the COSMIN guidelines for assessing the quality of structural validity.[18] Giannetta *et al* evaluated the cross-cultural validity and hypothesis testing as sufficient. The reliability was appraised as sufficient with the inter-rater agreement of 88%.[18] Giannetta *et al* did not report any analysis concerning measurement error, responsiveness and construct validity of MMD-HP. Finally, they judged the criterion validity as lacking.

### DISCUSSION
### Main findings

This study focused on identifying valid PROMs to assess MD in animal care workers. A two-step systematic review revealed the scarcity of standardised measures in this field. Only one specific measure for veterinarians, the MMD-AP, was identified and evaluated for content validity following COSMIN guidelines.[15 19] This finding was unexpected, particularly given the substantial number of studies on moral issues among animal care workers (ie, 455 records

in our databases). Striking also was the fact that very few studies used standardised questionnaires or PROM for the MD measurements, that is, the most important eligibility criteria in our systematic review. This points out a need to create awareness in this research field regarding the importance of a valid and reliable outcome measurement using high-quality, validated methods.

In our evaluation based on COSMIN guidelines, we found the quality of MMD-AP development to be doubtful, with inconsistent content validity and low-quality evidence.[20] 'Doubtful' ratings in our findings do not suggest an ordinal measure of PROM quality in its development; rather, they suggest lack of qualitative methods applied to develop and to evaluate PROMs, and poor reporting of application of these methods.

The findings highlight several critical issues. First, MD in animal care workers has not received adequate attention compared with other outcomes like anxiety, compassion fatigue or burnout. Second, many studies claiming to measure MD did not follow a valid or standardised methodology. Third, there seems to be a disregard for proper PROM development and psychometric validation methods in current research in this area. This gap in methodological rigour hinders understanding the nature, determinants and prevention of MD among animal care workers. However, this issue is not unique to animal care but is also prevalent in other fields, as demonstrated by the review of Giannetta *et al*.[18] Their review, included in the COSMIN database, did not assess the content validity of the reviewed PROMs as required by COSMIN guidelines, reflecting a broader trend of methodological shortcomings in current research practices.

The only PROM available for measuring MD in animal care workers is the MMD-AP.[15] Due to its current inconsistent content validity and low-quality evidence, further studies are necessary for its validation and improvement. Before its reassessment, the MMD-AP is not recommended for use in research or clinical practice, as it fails to meet COSMIN criteria of relevance, both comprehensiveness and comprehensibility.

Improving the quality of MMD-AP's content validity assessment requires qualitative studies involving the target population, including animal shelter workers, laboratory workers and veterinary nurses. This could result in

changes or the introduction of new items, leading to an updated version of the MMD-AP that would necessitate validation studies.

## Limitations and strengths

Certain limitations must be acknowledged, inherently dependent on the design and execution of this review.

First, COSMIN guidelines provide only a loose directory in cases where reporting of qualitative studies is poor. It gives only general rules for content validity studies, defining what must be reported. It does not provide guidance on how much detail is required to meet the criteria of the COSMIN methodology for assessing content validity.[31] Indeed, COSMIN provides guidelines for an indirect assessment of PROM quality by methodological quality of development and subsequent validation studies of PROMs. Thus, reporting of qualitative studies is one of the main indicators of methodological quality of PROMs.[21] Although COSMIN guidelines are precise which methodological steps must be reported, there is no guidance on how much details they should be. Due to this shortcoming, often qualitative studies are insufficiently reported and rated as 'doubtful'.[31]

Second, development studies of PROMs in this review suffer from novelty of COSMIN guidelines. While they started to appear in late 2010s, majority of original development of MD scales dates to early 2000s,[29 30] before the publication of these guidelines. This may explain the lack of qualitative studies and patient involvement. Yet, while we did see those qualitative methods being introduced for human healthcare professionals, these methods are lacking for veterinary medicine practitioners.

Lastly, MD has been coined 40 years ago in the nursing discipline and since then further definitions have been developed, possibly leading to a heterogeneous and complex understanding of MD. However, since COSMIN evaluation relies on a degree of subjectivity by reviewers to rate the standards of criteria,[20] not specific enough definitions and operationalisations may endanger evaluation of PROMs without additional qualitative studies.[32] For example, we encountered both moral stress and MD in our first review. Moreover, while we observed that there are many studies focusing on conceptual aspects of moral issues among animal care workers, very few studies used standardised questionnaires or PROM for the MD measurements. The use of standardised questionnaires was a criterion for the eligibility in this review. Hence, in the end, we could only identify and evaluate one specific scale for animal care workers. A recent study by Buchbinder et al[33] focused on differential assessment of moral stress and MD.[33] They discuss that moral stress is linked to systemic overburdening and does not necessarily involve feelings of powerlessness, contrary to MD. MD focuses more on specific stressors encountered in clinical settings than on the systemic causes of these stressors in healthcare. Therefore, it is simpler to create questions that reflect these clinical encounters when developing a standardised questionnaire. At the same time, current MD scales may suggest that MD is an individual pathology, rather than a complex syndrome, due to the lack of attention given to systematic constraints.

To deal with this subjective nature of rating, we relied on the transparent and systemic methodology of COSMIN.[20] The study followed PRISMA guidelines and pre-registered with PROSPERO. This ensures a thorough, transparent and traceable methodological approach. The adherence to COSMIN guidelines for assessing the content validity of PROMs ensures a rigorous evaluation of the measurement properties of outcome measures. Our evaluation conducted on every version of the scale; and informed by a clearly articulated clinical or research question(s), the methodological quality of PROM design, the development process, findings of the content validity study and content of the scale itself. By synthesising robust assessment of COSMIN framework with transparent methodological approach, this study primes the design and standardisation of valid MD assessments.

## Interpretations

An accurate assessment of a health problem in terms of incidence or prevalence requests a validated standard for measuring this problem in clinical, occupational or research settings. It could ideally be a validated diagnostic test, or a consensually approved set of diagnostic criteria, such as those used for quantifying the burden of major depressive disorder or general anxiety disorder in the global burden of diseases.[34] Unless such tests or criteria have been established, PROMs can help estimate the amount of the problem, as it was recently shown in occupational burnout research.[35] Yet, the prerequisite for this is an appropriate choice of the most valid PROM.[36 37] Compared with occupational burnout, MD is a relatively recent construct in occupational psychology; therefore, the research on MD could benefit from and capitalise on the methodological development in subjective outcome measurement in the other fields.

## CONCLUSION

This is the first systematic review focused on MD measurement in animal care workers. It showed that MD is rarely measured using standardised and evidence-based methods. These findings underline the need to develop and thoroughly validate such methods in the context of animal care. We think it is also necessary for the well-being of the animals, as we believe that MD has an impact on the animals cared for by the animal caretakers. More generally, methodological improvement and training in psychometry are recommended when developing and validating scales of MD and other subjective outcomes.

**Acknowledgements** The authors thank the Unisanté documentarist, Thomas Brauchly, for his precious help in establishing the literature search queries.

**Contributors** Conceptualisation: IGC. Methodology: NG, YB, IGC. Software: NG, YB. Validation: NG, YB, IGC, SH. Formal analysis: NG, YB. Investigation: NG, YB, IGC. Resources: NG, YB, IGC. Data curation: NG, YB, IGC. Writing—original draft preparation: NG, YB, IGC. Writing—review and editing: NG, YB, IGC, SH. Supervision:

IGC, SH. Project administration: IGC, SH. Funding acquisition: IGC, SH. Guarantor: IGC. All authors have read and agreed to the published version of the manuscript.

**Funding** This project has received funding from Swiss National Science Foundation (SNF) for the NRP79 project 407940_206396 'Linking animal and human welfare–refining rodent euthanasia'.

**Competing interests** None declared.

**Patient and public involvement** Patients and/or the public were not involved in the design, or conduct, or reporting, or dissemination plans of this research.

**Patient consent for publication** Not applicable.

**Provenance and peer review** Not commissioned; externally peer reviewed.

**Data availability statement** All data relevant to the study are included in the article or uploaded as supplementary information. Search syntax is available in PROSPERO (CRD42023422259).

**ORCID iDs**
Yigit Baysal http://orcid.org/0009-0003-6364-898X
Sonja Hartnack http://orcid.org/0000-0002-5757-5708
Irina Guseva Canu http://orcid.org/0000-0001-7059-8421

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
