## [Reviewer comments · BMJ Open]

ARTICLE DETAILS

TITLE (PROVISIONAL)	Moral distress measurement in animal care workers: a systematic review
AUTHORS	Baysal, Yigit; Goy, Nastassja; Hartnack, Sonja; Canu, Irina

VERSION 1 – REVIEW

REVIEWER	Reese, Laura A. MSU
REVIEW RETURNED	28-Nov-2023

GENERAL COMMENTS	I am in a bit of a quandary over this manuscript. On the one hand the analysis is careful, detailed, and sophisticated. On the other hand, it is focused on only two publications and essentially creates a straw man out of the paper applying the PROM to veterinary professionals. The systemic literature protocol is very well done and the fact that they found only a single paper applying PROM to veterinarians obviously makes the point that more work is needed in this area. But, given the dearth of work in this area, is it really surprising that that single development of a set of indicators has not been fully validated? I understand that according to the current paper, the earlier authors should have done more to validate the measures, but it is unclear whether they were actually not fully validated or if the earlier authors just did not fully explain their process. At the end of the day I have no particular criticism of what has been done in this manuscript, but it appears to be a case of form over substance and I am not sure about its contribution to the literature. This leaves me with nothing to say about suggested improvements so I will leave this up to the editors about the nature of the contribution here.
---

REVIEWER	Zimmermann, Claudia Medical University of Vienna, Department of Epidemiology
REVIEW RETURNED	12-Jan-2024

GENERAL COMMENTS	Thank you for the opportunity to review this interesting paper. This study examines the literature on moral distress measurements in animal care workers. The authors conducted a systematic literature review to find patient-reported outcome measures (PROMs) for this purpose, and consequently also assessed the psychometric properties of the PROMs currently in use. In my view, the topic is relevant and the chosen methodology is suitable, and the execution of the systematic review appears to conform with current standards. The obtained findings are interesting and the concluding remarks – that there are currently no validated measures for moral distress in animal workers
---

	recommended for use – is helpful information for the research community in this field. However, I believe that this report would greatly benefit from thorough revisions regarding these main aspects: Argumentation and references: In my view, the background does not quite provide a smooth introduction to the topic and the relevance of this study. More detail is needed to link concepts and ensure the reader is not left to put the pieces of evidence together by themselves. For example, it would be helpful to explicitly discuss the need for a specific animal care worker PROM and to explain why the existing ones for healthcare professionals are not sufficient. Some of the selected references do not seem to be exactly on topic, and others are missing (more comments below). The argumentation for methodological decisions often appears weak to me. For example, the explanation of step 2 in the review process does not really provide a convincing reason for the chosen course of action: the previous systematic review by Gianetta et al. did not appropriately assess content validity of 32 measures (no further information about these measures provided), so the authors “consequently” perform a content validity assessment for the one measure they found in their own search (step 1). Why? A more detailed explanation would be helpful, perhaps a better description of design and shortcomings of this existing review. Were there no PROMs among those 32 measures? What was the research question and the conclusion? This might be useful information to establish a better framework for the present study and to highlight its contribution, and also to make it easier for the reader to follow the deliberations of the authors. Presentation of results: The authors clearly state that only one measure, MDD-AP, met the inclusion criteria. Yet a substantial part of the Results is dedicated to the presentation and discussion of MDD-HP. Why the focus on the original MMD-HP by Epstein et al. when it is not targeted at animal care workers? There is no explanation for this choice, and even though MDD-HP features prominently in the Results it is completely absent from the Discussion. Similarly, why is the paper restricted to the MMD-AP for associates and does not include the one for owners as well? Since these are crucial decisions for what constitutes the outcomes of this review, there needs to be a stronger argument supporting these choices. There is another important aspect to the assessment of the selected PROMs that I am confused about. It is mentioned that the authors of the selected papers were contacted for more information, and yet the assessment reveals that information is lacking for several crucial points. Were the original authors unresponsive, unable to provide this information or is the assessment strictly based on the published reports? Language: The paper is mostly well-written, but there are several sentences with odd structure or unclear meaning that should be improved. Further proofreading, perhaps by a native English speaker, is recommended. Further comments to the authors Strengths and limitations 57-58: Rephrase for clarity. Background 95: I believe reference 7, while dealing with suicide, does not say anything about animal care workers. There are plenty of studies
--	---

	reporting a higher suicide risk for veterinarians especially, e.g. https://doi.org/10.2460/javma.255.5.595 (for the US), https://doi.org/10.1016/j.psychres.2023.115170 (Austria) 97: Please provide a reference for the 3R principles. 108: Please provide references for the systematic reviews. 119-120: Other than content validity being the most important psychometric property, are there any good reasons for solely focusing on content validity? Methods 128: I would appreciate more specific references about which COSMIN guideline, checklist etc. (they get mentioned further down in the Methods section but would be useful here as well). 130: I believe this reference (9) is not the correct one. 133-134: Consider rephrasing. 145-146: What was the reason for the chosen publication period of 1988 onwards, and why was this changed from the proposed period of 1973 onwards in the registration of the review? 150: The reference lists of which studies? All initial results or after first/second screening stage? 154-159: Please identify the reviewers in their various roles (e.g. by adding initials in brackets), the same applies to the process of PROM assessment (173-176). Results 191: Specify which inclusion criterion was not met by “only using the moral distress prevalence as one of the covariates” in this one article. 205-212: Please rephrase this paragraph, improve language and clarify your argument. 251-252: Which five components are you referring to? The ones by Epstein et al.? 252-258: Not sure what these examples are demonstrating here. Is this at all connected to the five components? 283-284: Unclear how overall assessment of quality was achieved. 300-301: Not sure why the authors specify that both reviewers rated something a certain way – was this the only time the reviewers agreed? Presumably not. Before presenting these results, there should be a comment on the extent of agreement between independent reviewer decisions. 322-323: Consider rephrasing, the meaning of this sentence is not quite clear. 327-328: The last sentence of this paragraph would be a better fit for the Discussion section. Discussion 353-354: This sentence should be rephrased for clarity.
--	--

VERSION 1 – AUTHOR RESPONSE

Reviewer #1

Dr. Laura A. Reese, MSU

1) I am in a bit of a quandary over this manuscript. On the one hand, the analysis is careful, detailed, and sophisticated. On the other hand, it is focused on only two publications and essentially creates a straw man out of the paper applying the PROM to veterinary professionals.

Response and performed amendment:

We agree with the reviewer’s statement (although we had hoped to find more publications on MD scales). Since we pre-registered the research plan of this review, we would like to stick to it, even with the low number of publications found. We also would like to raise awareness

that content validity is a crucial aspect – which we think is relevant for scales developed in future. Therefore, we kindly like to proceed with this 2-step analysis.

Performed amendment:

We revised the Discussion section Main Findings, (L. 384-389) and commented it as a study limitation (L. 435-451).

2) The systemic literature protocol is very well done and the fact that they found only a single paper applying PROM to veterinarians makes the point that more work is needed in this area. But, given the dearth of work in this area, is it really surprising that that single development of a set of indicators has not been fully validated? I understand that according to the current paper, the earlier authors should have done more to validate the measures, but it is unclear whether they were actually not fully validated or if the earlier authors just did not fully explain their process. At the end of the day, I have no particular criticism of what has been done in this manuscript, but it appears to be a case of form over substance, and I am not sure about its contribution to the literature. This leaves me with nothing to say about suggested improvements so I will leave this up to the editors about the nature of the contribution here.

Response:

We understand the Reviewer's feelings and share them.

Indeed, in the COSMIN framework, the validity assessment is based on a series of PROM development and content validity studies, where qualitative methods are given an important weight. This reflects a relatively recent trend to favor qualitative research in medicine, but also the ascent of clinometrics and psychometrics as a research field. As we commented in the revised Discussion, many PROMs' authors might be still unaware of these quite recently published guidelines (2018). For us it was also unclear if the validation was not done or just not reported. Therefore, we corresponded with the authors about reporting these details to see if these qualitative studies are reported elsewhere besides the included full texts. We accounted for the responses we received in our evaluation and mentioned them in the Discussion section Quality of psychometric validity of the selected PROMs (L. 309-312).

Here is the summary of our correspondence.

MMD-HP:

We asked whether they conducted further content validity studies on their instrument. The author answered that they did not. Still, it was clear from their publications that they conducted an elaborate face validity study with experts (Ethics Consultation Service of the University of Virginia) in a focus group study where they received feedback on item clarity and relevance of these items. We also asked about any records (e.g., verbatim transcriptions) kept from this study. The author answered that the keeper of these records, co-author Prof. Ann Hamric, passed away suddenly in 2019, and the records were lost.

MMD-AP:

We asked whether they conducted further content validity studies on their instrument. The author answered simply that they did not conduct any additional testing that was not reported in the study itself.

Reviewer #2

1) In my view, the background does not quite provide a smooth introduction to the topic and the relevance of this study. More detail is needed to link concepts and ensure the reader is not left to put the pieces of evidence together by themselves. For example, it would be helpful to explicitly discuss the need for a specific animal care worker PROM and to explain why the existing ones for healthcare professionals are not sufficient.

Response and performed amendment:

We elaborated in the amended Introduction (L. 80-105) on the unique stressors of animal care work, and why we think that moral distress phenomena in human health care may differ from animal care.

2) The argumentation for methodological decisions often appears weak to me. For example, the explanation of step 2 in the review process does not provide a convincing reason for the

chosen course of action: the previous systematic review by Gianetta et al. did not appropriately assess the content validity of 32 measures (no further information about these measures provided), so the authors “consequently” perform a content validity assessment for the one measure they found in their own search (step 1).

Response and performed amendment:

Response to course of action: In step 1 of the systematic review, we identified moral distress measures used in animal care worker studies. During this process, we screened all 560 studies that we exported from 3 databases. We encountered only MMD-AP (Kogan and Rishniw, 2023) that fulfilled our inclusion criteria. In its full text, we realized that MMD-AP is an adapted version of MMD-HP which is designed for healthcare professionals.

Before evaluating content validity in step 2, the user manual (Terwee et al. 2018; Mokkink et al. 2018) recommends searching systematic reviews of PROMs’s content validity in the COSMIN database. We found no systematic review for our target population, but we found the systematic review from Gianetta et al. By having a closer look, we realized that Gianetta et al. hadn’t assessed the validity (although it was described as if they had done it).

We had already foreseen to assess the validity according to Cosmin and provided this information in the pre-registration. In our approach, the Gianetta systematic review was an incidental finding, and we took the occasion to discuss this review in light with the human healthcare MMM-HP, which is the ancestor of the MMM-AP in our target population.

We revised the Methods section (L. 185-189) and the Results section (L. 223- 230) of the manuscript to clarify this strategy.

3)...Why? A more detailed explanation would be helpful, perhaps a better description of the design and shortcomings of this existing review. Were there no PROMs among those 32 measures? What was the research question and the conclusion? This might be useful information to establish a better framework for the present study and highlight its contribution and to make it easier for the reader to follow the deliberations of the authors.

Response:

Please see also our comments above. Our current systematic review and the previous systematic review by Gianetta et al. have certain similarities in their research questions, objectives, and study design as outlined below.

Research questions/objectives:

In both of our research questions/ objectives, we try to identify measures of moral distress and evaluate them according to the measurement properties checklist of COSMIN (Terwee et al. 2018; Mokkink et al. 2018). One important distinction is that while a systematic review by Gianetta et al. targets the population of human healthcare workers, ours targets animal care workers as a specified population. We both conducted our search strategy according to the Population, Exposure, Outcomes, or themes (PEO) framework (Bettany-Saltikov, 2012).

However, since we identified different target populations, we reasoned that exposures and outcomes may differ as well. For example, in our search strategy, we put search strings related to animal euthanasia and animal mercy killings for exposures; we also utilized recent surveys and cross-sectional study outcomes of moral distress among our target population as outcomes.

Study design:

Both reviews are conducted in a two-step systematic review procedure. In the first step, abstracts of imported studies from different databases are evaluated based on the inclusion/exclusion criteria. Then, full texts of included studies are again evaluated based on the same criteria.

The biggest difference between the two reviews is that Gianetta et al. do not undertake content validity evaluation which is the fundamental psychometric property. Content validity property suggests the ability of the PROMs to be an adequate reflection of the construct they measure by its content (Mokkink et al., 2010). Indeed, a PROM’s content must capture all the aspects of the construct it intends to measure in the target population and with their item selection in a given scale, amongst different populations. COSMIN content evaluation tracks

the appropriate methodological steps such as pilot studies with experts, cognitive interviews and focus groups with patients.

Performed amendment:

We elaborated on how we consider Gianetta et al.'s review in parallel to ours in the result section of the amended manuscript (L.218-229)

4) Presentation of results:

The authors clearly state that only one measure, MDD-AP, met the inclusion criteria. Yet a substantial part of the Results is dedicated to the presentation and discussion of MDD-HP. Why the focus on the original MMD-HP by Epstein et al. when it is not targeted at animal care workers? There is no explanation for this choice, and even though MDD-HP features prominently in the Results it is completely absent from the Discussion.

Response:

The main reason to present MMD-HP in addition to MMD-AP is that for the latter one, no information on development is given. We wanted to be fair, because we assumed that part of the information concerning the development could be obtained from the MMD-HP ancestor. Although, in cases when PROM is adapted to be used in another target population than the original target population for which it was developed, a new evaluation is necessary to appraise the relevancy of items in the original PROM (Terwee et al. 2018; Mokkink et al. 2018)).

To clarify we amended the text (L. 230-239).

5) Similarly, why is the paper restricted to the MMD-AP for associates and does not include the one for owners as well? Since these are crucial decisions for what constitutes the outcomes of this review, there needs to be a stronger argument supporting these choices.

Response and Performed amendment:

Since the associate's version contains three items more (the rest being identical to the owner's version) we chose to present only one detailed evaluation for the associates version. Furthermore, since content validity was not assessed, we cannot differentiate between the two target populations and decided to select the more comprehensive version.

The results section has been amended to clarification this choice (L. 241-247)

6) There is another important aspect to the assessment of the selected PROMs that I am confused about. It is mentioned that the authors of the selected papers were contacted for more information, and yet the assessment reveals that information is lacking for several crucial points. Were the original authors unresponsive, unable to provide this information or is the assessment strictly based on the published reports?

Response:

We did indeed correspond with the corresponding authors of the scales (Prof. Lori Kogan for MMD-AP; and Prof. Beth Epstein for MMD-HP). Here is the summary of our correspondence.

MMD-HP:

We asked whether they conducted further content validity studies on their instrument. The author answered that they did not. Still, it was clear from their publications that they conducted an elaborate face validity study with experts (Ethics Consultation Service of the University of Virginia) in a focus group study where they received feedback on item clarity and relevance of these items. We also asked about any records (e.g., verbatim transcriptions) kept from this study. The author answered that the keeper of these records, co-author Prof. Ann Hamric, passed away suddenly in 2019, and the records were lost.

MMD-AP:

We asked whether they conducted further content validity studies on their instrument. The author answered simply that they did not conduct any additional testing that was not reported in the study itself.

7) Language: The paper is mostly well-written, but there are several sentences with odd structure or unclear meanings that should be improved. Further proofreading, perhaps by a

native English speaker, is recommended.

Response and Performed amendment:

8) Strengths and limitations 57-58: Rephrase for clarity.

Response and Performed amendment:

The sentence has been rephrased for clarity (L. 57-58).

9) 95: I believe reference 7, while dealing with suicide, does not say anything about animal care workers. There are plenty of studies reporting a higher suicide risk for veterinarians especially, e.g. <https://doi.org/10.2460/javma.255.5.595> (for the US), <https://doi.org/10.1016/j.psychres.2023.115170> (Austria)

Response and Performed amendment:

Reference 7 is a cohort study on the working-age Swiss population where risk estimates of suicide mortality are conducted by Standardized mortality ratios (SMRs) for suicide by occupation and economic activity/industry stratified by gender. We put this reference because it shows high-risk estimates of suicide mortality among males. However, in retrospect, we agree with the reviewer that it is not the target population that we defined. We changed this reference to the recent Austrian study by Zimmermann et al. (2023) that the reviewer has provided. We thank the reviewer for their due diligence.

10) 97: Please provide a reference for the 3R principles.

Response and Performed amendment:

The reference for 3R principles has been provided (Grimm et al., 2023).

11) 108: Please provide references for the systematic reviews.

Response and Performed amendment:

Reference for systematic review has been provided (Gianetta et al., 2020).

12) 119-120: Other than content validity being the most important psychometric property, are there any good reasons for solely focusing on content validity?

Response:

The reasoning behind starting with content validity is that other measurement properties, such as construct validity or internal reliability, would be biased and incomplete without sufficient content validity (Mokkink et al., 2010; Rothmund et al., 2023). Since PROMs rely on reported outcomes of a given target population in their scale items, a degree of confidence is needed for whether the items comprehensive and relevant to reflect the theoretical foundation of the construct that the PROM is supposed to measure, degree of confidence for the content of the items in the PROM, any other kind of measurement properties will be potentially misleading (van Andel et al., 2020).

Performed amendment:

Method section Step 2. A systematic review of psychometric validity of the identified moral distress scales (L. 168-185) are amended to better elaborate on the COSMIN taxonomy of measurement properties and order of evaluating COSMIN measurement properties.

Methods

13) 128: I would appreciate more specific references about which COSMIN guideline, checklist, etc. (they get mentioned further down in the Methods section but would be useful here as well).

Performed amendment:

We added the following (Mokkink et al., 2010; Terwee et al., 2018; Mokkink et al., 2018).

14) 130: I believe this reference (9) is not the correct one. 133-134: Consider rephrasing.

Response:

This should be a mistype while managing references. Thanks for your diligence. We corrected this mistake in the performed amendment and rephrased it (L. 134).

15) 145-146: What was the reason for the chosen publication period of 1988 onwards, and why was this changed from the proposed period of 1973 onwards in the registration of the review?

Response:

During the study planning phase, we first thought to give an arbitrary but long period of 50 years of literature. Then, we encountered with first coinage of the term moral distress by Jameton dated back to 1984. Then, the publication period changed to 1984. It seems we made a mistake while preregistering and wrote 1988 instead of 1984. Thanks for your diligence.

Performed amendment:

We used three databases: PubMed, EMBASE, and PsycINFO to search for eligible studies published between January 1984 and April 2023.

16) 150: The reference lists of which studies? All initial results or after the first/second screening stage?

Response and Performed amendment:

Reference lists were manually checked to identify additional studies, during the second screening of full texts of included studies.

The methods section has been amended for this clarification (L. 154).

17) 154-159: Please identify the reviewers in their various roles (e.g., by adding initials in brackets), the same applies to the process of PROM assessment (L. 173-176).

Performed amendment:

Roles of reviewers are assigned to these parts and the Method and Result section is amended (L.156-165; 224).

Results

18) 191: Specify which inclusion criterion was not met by “only using the moral distress prevalence as one of the covariates” in this one article.

Response:

Our aims in this systematic review were to identify and evaluate measurement properties of moral distress scales included. To do so, we needed studies not just using moral distress scales to report the prevalence of moral distress; but also, they were needed to assess at least one of the psychometric properties mentioned in COSMIN guidelines (i.e., content validity, face validity, construct validity, structural validity hypothesis testing, cross-cultural validity, criterion validity, internal consistency, reliability, measurement error, responsiveness, interpretability) in their study population (Eligibility Criteria 2; L. 143-147).

19) 205-212: Please rephrase this paragraph, improve language, and clarify your argument.

Performed amendment:

The paragraph has been divided into two paragraphs and revised for clarification (L. 232-247).

251-252: Which five components are you referring to? The ones by Epstein et al.?

Response:

Yes, we are referring to the Epstein et al. citation mentioned earlier. These five components are also used in Kogan & Rishniw’s MMD-AP.

Performed amendment:

The result section has been amended to clarify the components and to give examples from items (L. 288-295).

20) 252:258: Not sure what these examples are demonstrating here. Is this at all connected to the five components?

Response:

These are some of the items of MMD-AP related to five components utilized both in MMDHP (Epstein et al., 2019) and MMD-AP (Kogan and Rishniw, 2023)

Performed amendment:

References are added to item examples (L. 291-296).

21) 283-284: Unclear how the overall assessment of quality was achieved.

Response and performed amendment:

The result section has been amended to clarify how the quality of evidence was achieved (L. 334-348)

22) 300-301: Not sure why the authors specify that both reviewers rated something a certain way – was this the only time the reviewers agreed? Presumably not. Before presenting these

results, there should be a comment on the extent of agreement between independent reviewer decisions.

Response and Performed amendment:

Argumentation on 'Doubtful' ratings of study quality due to poor reporting has been elaborated in the Discussion section (L. 391-394) and mentioned as a study limitation (L. 417-425)

23) 322-323: Consider rephrasing, the meaning of this sentence is not quite clear.

Performed amendment:

"Nonetheless, no confirmatory factor analysis was performed, contrary to the COSMIN guidelines for assessing the quality of structural validity." (L. 376-377)

24) 327-328: The last sentence of this paragraph would be a fitter fit for the Discussion section. Response and Performed amendment:

The last sentence has been removed from Results (L.382) and integrated as a study limitation (L.435-451)

25) 353-354: This sentence should be rephrased for clarity.

Performed amendment:

"This could result in changes or the introduction of new items, leading to an updated version of the MMD-AP that would necessitate validation studies." (L. 411-413)

VERSION 2 – REVIEW

REVIEWER	Zimmermann, Claudia Medical University of Vienna, Department of Epidemiology
REVIEW RETURNED	22-Mar-2024

GENERAL COMMENTS	In my view, the authors have satisfactorily addressed most of the issues raised and made substantial improvements to the paper. I have just a few minor points left (see attached file). In my view, the authors have satisfactorily addressed most of the issues raised and made substantial improvements to the paper. I have just a few minor points left. 1) 206-207: There seems to be a mix-up regarding the two versions of MDD-AP. Both here and in line 241 the authors mention "animal owners", which I believe is not correct since one version of the MDD-AP is targeted at owners of veterinary practices. It should be explained clearly when first mentioned (the Kogan study only looked at veterinarians and distinguished between owners/partners of a practice vs. associates) and then used consistently. I apologise for not pointing this out in the first round of comments, I had looked up the Kogan paper back then and only now realised that this distinction is not clear without consulting this additional reference. Sorry! 2) Original comment: 145-146: What was the reason for the chosen publication period of 1988 onwards, and why was this changed from the proposed period of 1973 onwards in the registration of the review? Authors Response: During the study planning phase, we first thought to give an arbitrary but long period of 50 years of literature. Then, we encountered with first coinage of the term moral distress by Jameton dated back to 1984. Then, the publication period changed to 1984. It seems we made a mistake while
--

	preregistering and wrote 1988 instead of 1984. Thanks for your diligence. Performed amendment: We used three databases: PubMed, EMBASE, and PsycINFO to search for eligible studies published between January 1984 and April 2023. The preregistration said 1973 and the first version of the paper said 1988, hence my question. This explanation about the concept origins makes perfect sense, perhaps it would be helpful to readers if you were to explicitly include this information in the Methods section (I know you mention moral distress being coined in 1984 in the strengths and limitations, but there is no connection to the time period for the search there). 3) Original comment: 300-301: Not sure why the authors specify that both reviewers rated something a certain way – was this the only time the reviewers agreed? Presumably not. Before presenting these results, there should be a comment on the extent of agreement between independent reviewer decisions. Authors Response and Performed amendment: Argumentation on ‘Doubtful’ ratings of study quality due to poor reporting has been elaborated in the Discussion section (L. 391-394) and mentioned as a study limitation (L. 417-425). I think there might have been a misunderstanding, I was trying to point out that the description of the content validity sounds odd when one sentence highlights that “both reviewers rated...” (as if the default would be that only one of them rates something a certain way). I would have liked to get some information on the agreement between the reviewers regarding the content validity assessment. Was there no disagreement at all? Did both reviewers have the same rating for all aspects? 4) Some language issues remain or have been introduced by adding new paragraphs (e.g. 244-246). I believe this could still be improved and some sections might benefit from streamlining for clarity, but I leave this to the judgement of the editor.
--	---

VERSION 2 – AUTHOR RESPONSE

1)

206-207: There seems to be a mix-up regarding the two versions of MDD-AP. Both here and in line 241 the authors mention “animal owners”, which I believe is not correct since one version of the MDD-AP is targeted at owners of veterinary practices. It should be explained clearly when first mentioned (the Kogan study only looked at veterinarians and distinguished between owners/partners of a practice vs. associates) and then used consistently.

I apologise for not pointing this out in the first round of comments, I had looked up the Kogan paper back then and only now realised that this distinction is not clear without consulting this additional

reference. Sorry!

Response:

We thank the reviewer for her diligence. In our previous, we amended this mistake of addressing practice owners/ associates as “animal owners” (page line. 243-244). However, it seems we did not amend consistently within full text.

Performed amendment: Page line. 209-210 has been amended.

2)

Original comment:

145-146: What was the reason for the chosen publication period of 1988 onwards, and why was this changed from the proposed period of 1973 onwards in the registration of the review?

Authors Response:

During the study planning phase, we first thought to give an arbitrary but long period of 50 years of literature. Then, we encountered with first coinage of the term moral distress by Jameton dated back to 1984. Then, the publication period changed to 1984. It seems we made a mistake while preregistering and wrote 1988 instead of 1984. Thanks for your diligence.

Performed amendment:

We used three databases: PubMed, EMBASE, and PsycINFO to search for eligible studies published between January 1984 and April 2023.

The preregistration said 1973 and the first version of the paper said 1988, hence my question. This explanation about the concept origins makes perfect sense, perhaps it would be helpful to readers if you were to explicitly include this information in the Methods section (I know you mention moral distress being coined in 1984 in the strengths and limitations, but there is no connection to the time period for the search there).

Performed amendment:

Page line 149-151 has been amended to connect the period decision with the coinage of the moral distress term.

“In the initial phase of planning our study, we considered reviewing literature over an arbitrary but long period of 50 years. However, upon discovering the term “moral distress” was first coined by Jameton in 1984, we adjusted our review period to start from that year onwards.”

3)

Original comment:

300-301: Not sure why the authors specify that both reviewers rated something a certain way – was this the only time the reviewers agreed? Presumably not. Before presenting these results, there should be a comment on the extent of agreement between independent reviewer decisions.

Authors Response and Performed amendment:

Argumentation on 'Doubtful' ratings of study quality due to poor reporting has been elaborated in the Discussion section (L. 391-394) and mentioned as a study limitation (L. 417-425).

I think there might have been a misunderstanding, I was trying to point out that the description of the content validity sounds odd when one sentence highlights that “both reviewers rated...” (as if the default would be that only one of them rates something a certain way). I would have liked to get some information on the agreement between the reviewers regarding the content validity assessment. Was there no disagreement at all? Did both reviewers have the same rating for all aspects?

Response:

Indeed, I [Y.B] and the other main reviewer [N.G] had disagreements in our individual ratings. Rayyan software allowed us to blind our ratings; so that, each reviewer can individually rate with no impression of the other. During our individual reviews, we scheduled regular meetings to discuss our disagreements with our supervisors [I.G.C and S.H]; so that, we can reach one final consensual rating (see page line 158-165). This methodology stems from COSMIN manual instructions.

The manual suggest that users of the COSMIN checklist undergo training and gain experience with the tool, complete it independently by at least two raters, and reach consensus on one final rating to improve the inter-rater agreement and reliability of the checklist evaluations.

We did not calculate any sort of agreement coefficient as the most important thing in the process is to get a consensual rating of the criteria and evidence.

Performed amendment:

Reference of the COSMIN guideline suggesting this consensual rating approach is cited in the page line 163-165.

4)

Some language issues remain or have been introduced by adding new paragraphs (e.g. 244-246). I

believe this could still be improved and some sections might benefit from streamlining for clarity, but I leave this to the judgement of the editor.

Response:

The amended new paragraphs are proof-red again, and some small changes are made in the full-text, focused on integrating the amended parts to the whole text.

VERSION 3 – REVIEW

REVIEWER	Zimmermann, Claudia Medical University of Vienna, Department of Epidemiology
REVIEW RETURNED	05-Apr-2024
GENERAL COMMENTS	Thank you for the opportunity to review this interesting paper, I have no further comments.